# EMS-effect of Exercises with Music on Fatness and Biomarkers of Obese Elderly Women

**DOI:** 10.3390/medicina56040158

**Published:** 2020-04-01

**Authors:** Jiyoun Kim, Yongseok Jee

**Affiliations:** 1Department of Exercise Rehabilitation and Welfare, Gachon University, Hombakmoero, Yeonsu-gu, Incheon 406-799, Korea; eve14jiyoun@gachon.ac.kr; 2Department of Leisure and Marine Sports, Hanseo University, Hanseo 1-Ro, Haemi-myeon, Seosan 31962, Korea

**Keywords:** exercises with music, cytokine, tumor necrosis factor, electromyostimulation, percent fat

## Abstract

*Background and objectives:* Electromyostimulation (EMS) has been shown to improve body composition, but what biomarkers it affects has not been investigated. The purpose of this study was to compare the EMS-effect of exercises with music on fatness and biomarker levels in obese elderly. *Materials and Methods:* Twenty-five women were randomly classified into a control group (CON) and EMS group (EMSG). EMS suits used in this study enabled the simultaneous activation of eight pairs with selectable intensities. Program sessions of EMS were combined with exercises of listening to music three times a week for eight weeks. Although both groups received the same program, CON did not receive electrical stimuli. *Results:* Compared with CON, a significant effect of the EMS intervention concerning decreased fatness, as well as an increased skeletal muscle mass and basal metabolic rate, were evident. Tumor necrosis factor-a, C-reactive protein, resistin, and carcinoembryonic antigen of biomarkers were significantly different in the groups by time interaction. Similarly, the positive changes caused by EMS were represented in lipoprotein-cholesterols. *Conclusions:* The results indicate that a significant effect due to the EMS intervention was found concerning body composition and biomarkers in obese elderly women.

## 1. Introduction

The elderly population is increasing over the world, and the incidence rate of chronic degenerative diseases is also increasing rapidly. According to a survey, 90% of seniors aged 65 and older have chronic diseases such as obesity, arthralgia, lower back pain, high blood pressure, and cancer, and more than half have difficulties in their daily lives [1]. Entering an aging society is a celebration of the increased life expectancy, but to be truly celebrated, chronic diseases must be solved; otherwise, the “old and sick population” will only increase and, as a result, the elderly will be a huge burden on society. The medical community and related officials think that there is an urgent need to establish a plan to sequentially reduce chronic degenerative diseases ahead of the “aging society”.

Among many chronic diseases, obesity causes complications, so there is an urgent need for a solution. In other words, obesity can get worse with age and, in older people, obesity is likely to cause more serious health complications, such as sarcopenia and loss of lean mass. Obesity is defined as a medical condition in which excessive amounts of body fat accumulate, resulting in detrimental health effects [2]. In women, obesity aggravates physical conditions [3], which are influenced by negative perceptions about loss of lean mass and body shape [2,4].

There are many methods for improving obesity, among them, regular exercise is regarded as an important means of relieving obesity. In other words, exercise is an important factor for increasing fat metabolism in the skeletal muscle, thereby causing fat oxidation. However, it is not easy for the elderly to participate in exercise like young people due to the degeneration of joints and muscles [4,5]. In terms of the relationship between degenerative arthritis and obesity, Kellgren [6] reported that obese people have a high incidence of degenerative arthritis in the knee and the first metatarsal bone. The reason for this is because the amount of fat in the lower limbs increases the distance between both femoral joints, and the knee joint becomes the genu varum. Taken together, previous studies have suggested that the risk factors for the development of the most prevalent degenerative arthritis among chronic diseases can be summarized as aging, being female, obesity, and sarcopenia. In patients with degenerative arthritis, rehabilitation exercise involves lower limb muscle development and weight loss through aerobic exercise [7,8]. Exercises mainly for weight loss include walking, biking, and swimming, which can reduce pain and improve physical ability. However, for elderly people who have degenerative arthritis and obesity, which promotes this disease, pursuing exercise to strengthen muscles without burdening the knee joint is recommended.

Recently, an advanced piece of equipment for improving body composition has been developed. Electromyostimulation (EMS) has the advantage of inducing muscle contractions without direct stimulation of the peripheral muscles by the central nervous system and providing similar effects to muscle contractions. In other words, EMS allows older people to reduce body fatness without having to provide excessive loads to muscles and joints. In particular, an obese person is likely to have problems with muscles and joints, so it is difficult to provide an excessive load, but EMS can provide effectiveness without causing such a problem. It is also gentle on the joints and reduces the risk of injury due to excessive loading [9]. EMS impulses are transmitted through electrodes on the skin located close to the dermis tissue for stimulation and modulate a variety of electrical wave forms, resulting in an electrical current that can be used to stimulate innervated muscles [10,11]. In the case of muscle contractions via EMS, the motor units under the control of the larger nerves are activated and muscle fibers with high thresholds are easily mobilized, resulting in positive effects on strength [12,13]. In the past, a researcher suggested that EMS breaks down the fatty capsule that covers the muscle and also improves the blood supply to the muscles, thereby helping to gain back lost muscle tone and return it to its original size [14]. In other words, it is effective in preventing obesity by reducing fat mass through EMS [15]. Until now, many researchers have reported that EMS has been used for the healing of pressure sores and improving muscular endurance and strength [16,17,18,19,20,21,22].

Although EMS has been reported to improve body composition, strength, and performance, there is a lack of research among obese elderly people. Moreover, few studies have been conducted on changes in cytokines related to obesity via EMS. While the degree of visual improvement in obesity is also important, the changes of blood biomarkers and lipids in the human body are more important. In particular, the aging process, when combined with obesity, contributes to increased cytokines and tumor factors generated by various organs, which may lead to health impairments or shorten one’s lifespan.

Therefore, the aim of this study was to investigate the changes in body composition and cytokines when EMS was used during the course of providing elderly people with exercises with music. This study examined a group of patients with obesity in a randomized controlled trial and assessed the physiological effects of the EMS intervention. The hypothesis of this study was divided into two as follows. First, the combination of exercises with music and EMS would bring about positive changes in body composition. Second, a combination of exercises with music and EMS would lead to positive changes in cytokines.

## 2. Materials and Methods

### 2.1. Study Design and Participants

This is a prospective, randomized, and controlled study, which takes EMS as the independent variable, and body composition and cytokines as the dependent variables. This study took place from 4 October to 4 December, 2018. The first assessment was conducted from 4 to 5 October, 2018 and the last assessment was conducted from 3 to 4 December, 2018. The participants in this study were all women, and those who applied to participate in the aerobic dance class. Most of the characteristics of the elderly were obese, and they also had joint disease, therefore, we tried to provide effective exercise to them.

The program period for exercises with music wearing an EMS suit was from 8 October to 30 November, 2018. Prior to the study, participants received detailed explanations regarding study procedures and were then asked to complete a questionnaire, then, selected subjects were to randomly classify the elderly who applied to enter exercises with music. The inclusion criteria required that patients were obese in terms of the percent of fat and had not exercised regularly for over six months. Additionally, patients were also included if they had not received treatment or medication for weight loss or anything known to affect body composition and cytokines, and if they did not have any internal metallic materials. Exclusion criteria consisted of having a history of impairment of a major organ system or a psychological disorder.

All patients knew they had to wear their EMS suits when exercising while listening to music, but they did not know if electrical stimulation was provided. In other words, they were assigned using random number tables and assigned identification numbers upon recruitment. In order to prevent communication between the electromyostimulation group (EMSG) and control group (CON) who was not provided with electrical stimulation, the patients were classified according to their community areas, and EMSG was sent to the center in the morning and CON in the afternoon. At the beginning of the measurement, only EMSG realized that a current was coming from their suits.

After excluding two patients out of thirty-two eligible participants, the remaining thirty patients belonged to one of two groups. Of the 15 patients in the CON who were allocated to the non-EMS group, one did not receive assessment and two were lost in the follow-up phase. Therefore, 12 patients in the CON were analyzed in our study. Furthermore, of the 15 patients in the EMSG, one did not receive assessment, and another was lost in the follow-up phase. Therefore, 13 patients of the EMSG were analyzed in our study as shown in Figure 1. Participant characteristics, which indicated homogeneity, are presented in Table 1.

### 2.2. Research Ethics

This study was conducted in accordance with the Declaration of Helsinki and was approved by the ethics committee (1 September 2018 to 31 August 2019; 2-1040781-AB-N-01-2017083HR). Written informed consent was obtained before enrollment. First, all of the patients arrived at Songdo hospital to sign an informed consent form and complete a self-reported questionnaire about their health status. After this procedure, all subjects participated in the experiment conducted by an expert.

### 2.3. Anthropometric Measurements

To measure body composition, all patients were weighed while wearing light clothes and no shoes. The bioelectrical impedance analysis was employed, using the BMS 330 for height and InBody 320 for body composition (Biospace Co., Ltd., Seoul, Korea). This analyzer is a segmental impedance device measuring voltage drops in the upper and lower body. Eight tactile electrodes located in palms, fingers, front soles, and rear soles were placed on the surfaces of the hands and feet. The precision of the repeated measurements expressed as the coefficient of variation was, on average, 0.6% for the percentage of fat mass [22]. This analyzer is a segmental impedance device in which the electrodes are made of stainless-steel interfaces. The subjects stood upright by placing their bare feet on the foot electrodes and gripping the hand electrodes. For accurate inspection, food intake and water intake were prohibited for 4 h and for 1 h before the test. In particular, urine, which may affect body weight and body fluid, was expelled 30 min before the test [23].

### 2.4. Biomarker Measurements

Blood samples were taken after fasting for 10 h or longer before assessment and were collected using BD vacutainer tubes (Becton Dickinson, Franklin Lakes, NJ, USA) at 8 am the following day. After the subjects were stabilized for 10–15 min, 5 mL of blood was collected from the antecubital vein of the subjects with a disposable syringe by a medical laboratory technologist before and after the experiments. A total of 2 mL of the 5 mL of venous blood was added to an anticoagulant tube (EDTA bottle), shaken, and centrifuged at 3000 rpm for 5 min. The remaining 3 mL was left at room temperature for 1 h and centrifuged at 1000 rpm for 15 min. Isolated plasma and serum were kept frozen until the test. The samples were taken to the laboratory for analysis, as follows.

Interleukin-6 (IL-6) from the serum was analyzed using an enzyme-linked immunospecific assay (ELISA) kit (Cohesion Biosciences, London, UK). The minimum detectable dose of Human IL-6 is typically less than 1 pg/mL. The Human IL-6 ELISA Kit allows for the detection and quantification of endogenous levels of natural and/or recombinant Human IL-6 proteins within the range of 3.9 to 250 pg/mL. Tumor necrosis factor-alpha (TNF-a) from serum was analyzed using an enzyme-linked immunospecific assay (ELISA) kit (Cohesion Biosciences, London, UK). The serum was allowed to clot in a serum separator tube at room temperature, was centrifuged at approximately 1000× *g* for 15 min, and was immediately analyzed [24]. The minimum detectable dose of Human TNF-a is typically less than 7 pg/mL. The Human TNF-a ELISA Kit allows for the detection and quantification of endogenous levels of natural and/or recombinant Human TNF-a proteins within the range of 15.6 to 1000 pg/mL. C-reactive protein (CRP) from the serum was also analyzed using an ELISA kit (Cohesion Biosciences). The minimum detectable dose of Human CRP is typically less than 10 pg/mL. The Human CRP ELISA Kit allows for the detection and quantification of endogenous levels of natural and/or recombinant Human CRP proteins within the range of 15.6 to 1000 pg/mL. Resistin (RSTN)—known as adipose tissue-specific secretory factor—from the serum was analyzed using an ELISA kit (Phoenix Pharmaceuticals, London, UK). The standard solution and sample for RSTN were added to a microplate coated with a specific RSTN monoclonal antibody and bound to RSTN (#EK-S-028-36) to form immobilized antibodies. Afterwards, unbound material was removed by washing and a biotinylated polyclonal antibody specific for biomarkers was added to each well. Unbound antibody-enzyme for biomarkers was removed by washing and Horseradish Peroxidase was added to each well, respectively. The carcinoembryonic antigen (CEA) was measured by the Sandwich principle of quantitative chemiluminescence assay, with a Cobas 8000 e801 (Roche Diagnostics, Mannheim, Germany). For reference, a normal value of CEA is ≦3.8 ng/mL for nonsmokers and ≦5.5 ng/mL for smokers. A binding substance was developed after the washing process and addition of substrate solution (TMB) [15]. The creatine kinase (CK) was also included because it was considered to be an indicator of muscle damage during or after exercise with music. The CK was analyzed using the Beckmann Coulter Inc. device (Brea, CA, USA) before and after the experiments [25]. For a normal value of CK at rest in a healthy adult, reference ranges are as follows: 52–520 IU/L for the high CK level; 35–345 IU/L for the intermediate CK level; and 25–145 IU/L for the low CK level [26]. Generally, high-density lipoprotein-cholesterol (HDL-C) and low-density lipoprotein-cholesterol (LDL-C) were measured by the homogeneous enzymatic colorimetric assay, with a Cobas C702 (Roche Diagnostics). For reference, a normal value of HDL-C is ≥40 mg/dL and a normal range of LDL-C is ≦100 to 129 mg/dL.

### 2.5. Calorie Intake/Output, Daily Physical Activity, and Working Capacity

This study investigated the calorie intake/output and daily physical activity in order to control and minimize the extrinsic variables that may affect the results of the experiment. Above all, the working capacity or maximum oxygen uptake was measured by exercise test to know the limit of physical activity of elderly women. Prior to the experiment, the obese patients were provided a diary to record what they consumed for a day throughout the experimental period. During this time, we calculated their daily caloric intake volume using CAN-Pro 5.0 (http://canpro5.kns.or.kr; Korean Nutrition Society, Korea) every day for eight weeks. The daily amount of physical activity that was performed outside the experiment was recorded and calculated. The patients answered the questionnaires based on the recordings of physical activity for the past seven days for eight weeks. The total score was obtained through the summation of the duration (in min) and frequency (days) of walking, moderate-intensity activity, and vigorous-intensity activity. Then, the data were used to calculate the amount of physical activity based on the international physical activity score conversion method using the metabolic equivalent (MET)-minutes score, as shown in Table 2 [27]. Finally, the calorie intake, calorie output, and physical activity were recorded, and the mean values of the above variables for four weeks were analyzed.

Maximum oxygen intake was measured only once before the experiment to determine the indicator that would confirm the aerobic nature of the exercises performed during exercises with music. A gas analyzer (Qurak CPET^®^, Cosmed, Rome, Italy), an electrocardiogram (ECG) analyzer (Heartwave II^®^, Cambridge Heart Inc., Tewksbury, MA, USA), and a treadmill ergometer (T150, HP/Cosmos^®^, Bavaria, Germany) were used for this experiment. For investigating the working capacity of elderly women, the modified Bruce protocol was inputted for the graded exercise test (GXT) and the speed and gradient controls were checked on the treadmill. All patients were restrained from vigorous physical activity and taking medication 48 h prior to the test and eating 3 h prior to the test. The electrodes were attached to the chest and the blood pressure cuff was placed on the brachial artery. The mouthpiece was fixed over the lip and nose area to breath only through the mouth and to the mouthpiece. In detail, the GXT of this study was to investigate coronary artery disease and/or abnormal rhythms and to evaluate exercise capacity. The modified Bruce protocol was applied in consideration of the elderly obese women. In the protocol, the speed stayed constant for the first three stages, starting at 1.7 mph at 0% incline. After the third stage, the speed and grade increased by 2.5 mph and 12%, respectively. After that, both the speed and grade increased every 3 min. All subjects were instructed to continue to walk or jog until reaching an all-out level which is their maximal ratings perceived exertion (RPE). During and after walking or running on the treadmill for as long as possible, the subjects were instructed to describe their symptoms as follows: chest pain, shortness of breath, dizziness, and leg pain. During the test, subjects were asked to express their level of exercise intensity on the RPE scale. The test was terminated if the following symptoms occurred: (a) drop in systolic blood pressure of more than 10 mmHg from baseline, despite an increase in workload, when accompanied by other evidence of ischemia; (b) moderate-to-severe angina; (c) increase in nervous system symptoms; (d) signs of cyanosis; (e) technical difficulties in monitoring electrocardiographic tracings; (f) subject’s desire to stop; (g) sustained ventricular tachycardia; and (h) ST elevation (>1 mm) in leads without diagnostic Q waves (other than V1 or a VR) [28]. The variable of this study was limited oxygen uptake calculated by body weight every minute at each test stage.

### 2.6. Exercises with Electromyostimulation Administration

All patients completed a supervised progressive program for eight weeks. Participants were given variously sized EMS suits made by Miracle^®^ (Seoul, Korea) according to their size. The suits were composed of a silicone conductive pad and wireless materials. The electrical strength of the suit was controlled via Bluetooth. EMS suits used in this study enabled the simultaneous activation of eight pairs of muscle groups (both upper legs, both upper arms, buttocks, abdomen, chest, lower back, upper back, and latissimus dorsi) with selectable intensities for each region. Based on recommendations from available literature [9,21,29,30,31], the stimulation frequency was selected at 85 Hz, the impulse-width at 350 μs, the impulse-rise as a rectangular application, and variable electrostimulation intensities relative to the maximum peak voltage. This study used 1 MT (maximal tolerance) as the maximum peak voltage, similar to calculating the maximal voluntary contraction as one maximal repetition [13]. Each 1 MT of the upper and lower body was measured and stored in Bluetooth, and the intensity was adjusted for each individual during aerobic dancing. In order to prevent the patients from being surprised or uncomfortable with the electrical stimulus, the 1 MT level was gradually increased while providing a low stimulation current [18,32,33,34]. The electric stimulation was stopped at the request of the participant when reaching an unbearable level on RPE scale [35], at which point the intensity was set as 1 MT. In other words, % MT of this study was obtained through RPE scale—which was a numerical scale that ranged from 6 to 20, where 6 means "no exertion at all" and 20 means maximal exertion. The intensity of exercise was estimated by applying RPE when exercising with music to CON as well as EMSG. The intensity of the electrical workout was different from 1 MT. The patients of EMSG were assigned to 60% of 1 MT from the baseline to Week 2, 70% of 1 MT from Week 3 to Week 5, and 80% of 1 MT from Week 6 to Week 8. Although the patients in CON performed exercises with music while wearing EMS suits, they did not receive any electrical stimuli. All patients were asked to express the difficulty level of exercise during exercises with music wearing an EMS suit [15]. An instructor asked for the RPE every 5 min, and an assistant recorded them. Meanwhile, the impulse duration was 6 s, with a 4-s break between impulses. For EMSG, an instructor conducted three times a week on two nonconsecutive days (Monday, Wednesday, and Friday) to allow for a rest interval of 48 h between each session.

In order to provide effective muscular contractions and to prevent harmful joint injury, the dance movements were simplified and composed of clapping and tapping, bending and rotating, aerobic and anaerobic exercises—which mean the aerobic exercise was the motion for 40 min dance and the anaerobic exercise was the muscular contraction for 6 second’s moving motion from EMS—and stretching exercises. The exercises with music wearing an EMS suit was as follows. Warm-up was performed at RPE of 9–11 for 5 min of walking in place. The upper and lower leg stretching was performed until they felt mild discomfort. Work-out with dance consisted of clapping and tapping (8 min), bending and rotating (12 min), aerobic and anaerobic exercises (10 min), and stretching exercises (10 min). Initially, clapping and tapping with dance began with applause for 1 min, followed by tapping of the head, shoulders, torso, back, and legs. At this time, the exercise intensity was between RPE 9 and 11. Second, bending and rotating with dance was performed between RPE 11 and 13 for 12 min with kneeling, followed by standing, standing on toes, waving arms, lifting legs, bending the waist upward, and rotating shoulders. Third, aerobic and anaerobic exercises with dance were performed between RPE 11 and 13 for 10 min with stepping forward and sideways, stepping backward and sideways, heel raises, lifting knees, curling legs, front lunges, and cross lunges. Fourth, stretching exercises with dance began with clapping and drawing an X-shape, followed by 10 min of RPE 11–15 for rotating arms, stretching arms, rotating wrists, shaking wrists, bowing, and rotating shoulders. Finally, the program was finished by stretching the upper and lower bodies for 5 min. Specifically, according to the investigation in this study, a combination of voluntary movement plus the evoked contraction during exercise tended to decrease by about 30% compared to the original range of motion (ROM) operation.

### 2.7. Data Analysis

All data were reported as mean ± standard deviation (SD), and these data were checked for normality distribution using Shapiro–Wilk’s W-test in SPSS 18.0 (SPSS Inc., Chicago, IL, USA) for Windows. Prior to analysis, we observed the difference between groups through Mann–Whitney U test before comparing between groups and times, as shown in Table 1. An analysis of variance (ANOVA) test was used to evaluate the significance of the differences between groups at baseline. Then, the effects of the interventions were assessed using an analysis of variance for repeated (2 × 2) measures (group, time, and group by time interaction). An intention-to-treat analysis was performed to compare the intervention group (EMSG) with the CON. The between-group factor was the study groups (i.e., CON vs. EMSG) and the within-group factor was the week (i.e., Week 0 vs. Week 8). The level of statistical significance chosen was *p* ≤ 0.05.

## 3. Results

### 3.1. Comparison of Demographics, Calorie Intake/Output, Physical Activity, and Working Capacity

There were no significant differences between groups for all variables as shown in Table 1. The maximum oxygen uptakes of CON and EMSG were 23.13 ± 4.25 mL/kg/min and 25.34 ± 6.51 mL/kg/min, respectively, and there was no significant difference between groups. In particular, percent fat was not significantly different between groups. As shown in Table 3, there were no significant differences between groups in calorie intake, calorie output, and physical activity level for the recorded week during the 8-week experimental period.

### 3.2. Effect of Electromyostimulation on Body Composition

As shown in Table 4, no significant effect of the EMS intervention was found in body weight when comparing the intervention and the control group. However, skeletal muscle mass (*F* = 7.826), fat mass (*F* = 8.717), percent fat (*F* = 4.961), and basal metabolic rate (BMR) (*F* = 28.770) were significantly different in group by time interaction. This result showed a significant effect of the EMS intervention concerning body composition was evident.

### 3.3. Effect of Electromyostimulation on Biomarkers

As shown in Table 5, the level of IL-6 in the CON showed increasing tendency from Week 0 to Week 8, but this level showed decreasing tendency in the EMSG, although with no significant difference between groups. However, TNF-a (*F* = 21.003), CRP (*F* = 27.825), RSTN (*F* = 9.520), and CEA (*F* = 19.331) were significantly different in group by time interaction. These positive changes were also represented in HDL-C and LDL-C. Specifically, although the level of CK in both groups showed increasing tendencies after the experiment, no significant difference was observed between groups. These results demonstrated that significant effects due to the EMS intervention were found concerning biomarkers in obese elderly women.

## 4. Discussion

This study found some evidence that exercises with music wearing EMS suits improved body composition. All variables in EMSG were significantly changed except for body weight. Additionally, although almost biomarkers in CON showed no change after eight weeks, the cytokines significantly improved in EMSG except for IL-6. Specifically, the CK—which is an indicator of muscle damage during or after exercise—between CON and EMSG was not significantly different by the end of the experiment. Characteristically, it was found that the exercises with music worn with EMS suit, which was envisioned and applied in this study, did not cause muscle damage to obese elderly women, as CK did not exceed the abnormal ranges (≥520 IU/L for the high CK level; ≥345 IU/L for the intermediate CK level) [26] before and after the experiment.

The results of this study showed that the oxidative effect of exercises with music combined with the muscle contractions provided by EMS had a beneficial effect. Moreover, in the obese elderly, it is in a situation where a lot of restrictions such as reduction of ROM can occur. In this case, the effect of exercise may be reduced due to the inability to effectively cause muscle contraction during exercise. At this time, EMS training induces more muscle contraction in most exercise movements, thereby giving positive effects to the patients. In other words, this positive effect in EMSG was thought to be crucial in the increase of skeletal muscle mass and BMR after the end of the experiment. The results of this study are thought to be similar to those of several research studies. A study reported that the improvement in anthropometric measures was greater for walking and EMSG compared with walking-only and CON in sedentary adult women after 8 weeks [36]. Another study indicated that there was a significant reduction in body fat when exercise was combined with EMS [37]. In fact, exercise with music is a type of oxidative exercise that can be enjoyed while listening to music. It can also develop the cardiopulmonary function, as well as muscular strength and endurance. However, the exercises with music used in this study were designed to avoid using the full range of motion to prevent joint complications in elderly women. For this reason, we think that the body composition and cytokines in the CON did not change in almost variables, compared with those of EMSG. In other words, CON was not positively altered in terms of body composition and cytokines due to the lack of EMS’s supplementary muscle stimulation. In particular, the biomarkers in CON showed significant increases or no changes, despite participating in exercises with music. Specifically, the TNF-a, CRP, and RSTN in CON significantly increased after eight weeks. However, these variables in EMSG significantly tended to decrease. These results revealed significant differences in group × time interaction. Meanwhile, the reason that there was negative or no change in the variables of CON is thought to be due to a small range of dancing activities for avoiding the severe change of degenerative joints. Moreover, only eight weeks of exercises with music in the elderly obese women is quite short, suggesting no changes in body composition or cytokines.

When screening any cancer, the CEA is widely investigated [38,39,40]. CEA, an oncofetal glycoprotein, is overexpressed in adenocarcinomas and is thus widely used as a tumor marker. CEA may be involved in the release of proinflammatory cytokines, probably by stimulating monocytes and macrophages [38], and in the release of endothelial adhesion molecules [36]. Therefore, CEA may contribute to the development of cancer. In addition, this action of CEA may also cause atherosclerosis and cardiovascular disease, as well as the metastasis of malignant cells [41,42]. CEA, for which aging is a major contributing factor, showed an increasing tendency in the CON of this study over the course of the 8-week program. Lee et al. [43] reported that CEA concentrations could be associated with metabolic disturbances and cardiovascular disease, as well as cancer. Several types of diseases are linked to higher levels of the CEA of biomarker components [41,42,44,45]. IL-6 also has many roles essential to the regulation of the immune response, hematopoiesis, and bone resorption. It is involved not only in the hepatic acute-phase response, but also in adipose tissue metabolism and lipoprotein lipase activity [46]. The overproduction of IL-6, a proinflammatory cytokine, is associated with a spectrum of age-related conditions, including cardiovascular disease, osteoporosis, arthritis, type two diabetes mellitus, certain cancers, periodontal disease, frailty, and functional decline [47]. Meanwhile, CRP is a major acute-phase reactant primarily synthesized in the liver hepatocytes. It is composed of five identical, 21,500-molecular-weight subunits. CRP mediates activities associated with preimmune nonspecific host resistance. It shows the strongest association with cardiovascular events.

It is no exaggeration to say that most of the diseases associated with the aging process, such as obesity or sarcopenia, are associated with vascular disease, cardiovascular disease, and tumorigenic diseases. Such chronic degenerative diseases can be prevented and treated by medication, surgery, and healthy lifestyles, but above all, it has been reported that exercise habits and regular exercise are more necessary for sustaining a healthy life [24]. Regular physical activity for elderly people can help them to maintain or sustain a healthy body weight, enhance muscle mass, and strengthen their immune system. According to the above theories, Rogers et al. [48] and McTiernan [49] showed that the modulation of energy balance by increasing physical activity can contribute to the reduction of cancer risks through numerous epidemiological reviews. Kobayashi et al. [50] also suggested that high levels of moderate and vigorous physical activity during adolescence may contribute to a lower risk of breast cancer in both pre- and post-menopausal women. Therefore, an emerging body of evidence suggests a strong inverse association between higher levels of physical fitness or greater amounts of exercise and cancer occurrence or mortality [51]. However, since there are some discrepancies regarding the exercise volume (intensity, time, and frequency) required for the prevention of cancer [24], more specific research is needed.

Many studies have reported that exercise can inhibit cancer development through the enhancement of immunity or promote cancer through the suppression of immunity [50,52]. Banerjee et al. [53] suggested that EMS was capable of eliciting a cardiovascular exercise response without loading the limbs or joints and inducing rhythmical contractions in the leg muscles. According to their results, they demonstrated significant improvements in peak oxygen consumption, walking distance, and quadricep strength after 6 weeks. In their findings, the EMS was only attached to specific parts of the body and the tolerance strength was only about 50%. In other words, most subjects in their study selected an impulse intensity that was consistent with the lower end of the training intensity zone. It is important that the exercise intensity is high enough to improve body composition and produce benefits of metabolism-related exercises [54]. This study provided exercises with music wearing an EMS suit, which used progressive impulse intensities. The use of EMS has been reported to be an effective complementary method to conventional exercise programs [55,56]. Although the favorable effects of EMS on neuromuscular parameters have been previously shown in elderly subjects [57,58], the effects of EMS on body composition and cytokines in elderly obese women are confirmed by the results of this study. Several studies have found that although the various impulse intensities of EMS showed improved tendencies in adipokine profiles, there was an increased effectiveness when using higher electrical impulse intensities [59,60]. One researcher reported that adipokines fluctuated irregularly when low and moderate EMS impulse intensities were applied during the experimental period. However, RSTN decreased regularly and sequentially for 6 weeks when a high EMS impulse intensity was applied [21]. The results of other research studies also show a decrease in RSTN due to exercise [15,61,62,63,64]. In other words, the stimulation [65,66] that was applied in this study with impulses greater than a moderate intensity can be regarded as an exercise stimulus that activates lipid metabolism, which the levels of HDL-C and/or LDL-C tended to increase and/or decrease in the EMSG; whereas the opposite tendencies appeared in those of the CON. Previous researchers have also reported that a higher level of physical activity is associated with an improved lipoprotein profile and increased fat oxidation [66]. Considering the results of this study, the effects of EMS with exercise while listening to music performed for 40 min per session with low to high impulse intensities provided benefits at the end of the experiment.

Ultimately, we suggest that exercises with music available for EMS suits can contribute to oxidative fat metabolism and cytokine reduction by effectively stimulating the muscles of obese elderly women, who may be less active or limited by a small range of motion. In other words, the effects of the progressive application of EMS intensities in this study were similar to the results of research [67] on endurance exercise training that showed increased lipid oxidation leading to positive effects on cytokines related to inflammatory substances and metabolic indicators, as well as body composition, in obese women.

## 5. Conclusions and Limitation

Based on the confirmed homogeneity of this study, the results suggest that the progressive electrical impulse of EMS for eight weeks may improve body composition and tumor- or inflammation-related cytokines in elderly obese women. However, since IL-6 of biochemical variables did not show positive changes in the EMSG, we suggest that the long-term exercises with music wearing an EMS suit should be used to find changes in body composition and biomarkers. In particular, in order to observe changes in biomarkers that are harmful to health, it is necessary to conduct research for a longer experimental period or greater exercise frequency. Although this study showed positive results in body composition and biomarkers, the sample size and larger trials are strongly needed to obtain excellent results.

## Figures and Tables

**Figure 1 medicina-56-00158-f001:**
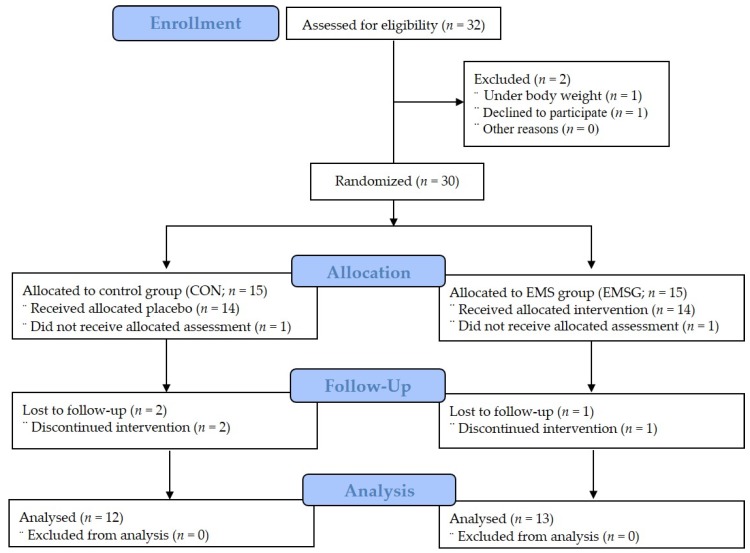
Patients’ allocation (consolidated standards for reporting of trials flow diagram). CON: Control group; EMS: Electromyostimulation; EMSG: EMS group

**Table 1 medicina-56-00158-t001:** Physical characteristics of the subjects.

Variables (Unit)	Groups		Mann–Whitney U Test	
	CON (*n* = 12)	EMSG (*n* = 13)	Z	*p*
Age (y)	71.75 ± 2.73	70.38 ± 2.93	−0.908	0.376
Height (cm)	151.49 ± 2.50	151.51 ± 4.02	−0.328	0.769
Weight (kg)	63.75 ± 3.82	63.05 ± 8.78	−0.272	0.810
Percent fat (%)	38.33 ± 4.31	39.21 ± 1.76	−0.082	0.936
VO_2_max (mL/kg/min)	23.13 ± 4.25	25.34 ± 6.51	−0.412	0.675

All data represents the mean ± standard deviation. VO_2_max: maximum oxygen capacity.

**Table 2 medicina-56-00158-t002:** Degrees of category scores by international physical activity questionnaire.

#	Activity Degrees	Criteria
1	Low	• No activity is reported or • Some activity is reported but not enough to meet #2 or #3.
2	Moderate	Either of the following three criteria as below: • 3 or more days of vigorous activity of at least 20 min per day or • 5 or more days of moderate-intensity activity and/or walking of at least 30 min per day or • 5 or more days of any combination of walking, moderate-intensity, or vigorous-intensity.
3	High	Any one of the following two criteria as below: • Vigorous-intensity activity on at least 3 days and accumulating at least 1500 MET-min/week or • 7 or more days of any combination of walking, moderate-, or vigorous-intensity activities accumulating at least 3000 MET-min/week.

Equations for calculating physical activity degree as follows; Walking metabolic equivalent (MET)-min/week = 3.3 × min of activity/a day × days per week. Moderate-intensity physical activity MET-min/week = 4.0 × min of activity/a day × days per week. Vigorous-intensity physical activity MET-min/week = 8.0 × min of activity/a day × days per week. Total MET-min/week = Walking MET-min/week + Moderate-intensity physical activity MET-min/week + Vigorous-intensity physical activity MET-min/week.

**Table 3 medicina-56-00158-t003:** Differences of controlled variables.

Items	Week	Groups		ANOVA (*p*)		
		CON (*n* = 12)	EMSG (*n* = 13)	G	T	G × T
Calorie intake	4	1663.21 ± 124.38	1671.36 ± 182.25	0.819	0.867	0.784
(kcal)	8	1688.69 ± 119.85	1641.43 ± 193.62			
Calorie output	4	246.25 ± 51.23	258.08 ± 49.25	0.910	0.442	0.919
(kcal)	8	229.97 ± 52.36	237.88 ± 51.69			
PAC	4	1.85 ± 1.45	1.83 ± 1.47	0.862	0.422	0.788
scores	8	1.84 ± 1.63	1.87 ± 1.78			

All values are expressed as mean ± standard deviation. All groups are scored by low-, moderate-, and high-activity levels. PAC: Physical activity category.

**Table 4 medicina-56-00158-t004:** Differences and changes in body composition.

Items		Groups		ANOVA (*p*)		
		CON (*n* = 12)	EMSG (*n* = 13)	G	T	G × T
Body weight	Pre	63.75 ± 3.82	63.05 ± 8.78	0.517	0.010	0.297
(kg)	Post	62.33 ± 4.17	57.54 ± 6.58			
Skeletal muscle	Pre	20.46 ± 0.97	20.94 ± 2.03	0.020	0.285	0.010
(kg)	Post	19.82 ± 1.71	22.41 ± 2.19			
Fat mass	Pre	25.23 ± 2.93	26.50 ± 5.50	0.649	0.014	0.007
(kg)	Post	25.42 ± 2.66	22.31 ± 3.89			
Percent fat	Pre	38.33 ± 4.31	39.21 ± 1.76	0.273	0.466	0.036
(%)	Post	39.66 ± 2.32	34.95 ± 3.82			
Basal Metabolic	Pre	1187.00 ± 35.92	1188.69 ± 67.98	0.004	0.249	0.001
Rate (kcal)	Post	1110.17 ± 40.62	1237.77 ± 74.40			

All data represents the mean ± standard deviation.

**Table 5 medicina-56-00158-t005:** Differences and changes in biomarkers.

Items (Units)		Groups		ANOVA (*p*)		
		CON (*n* = 12)	EMSG (*n* = 13)	G	T	G × T
IL-6	Pre	14.29 ± 7.52	14.51 ± 7.14	0.236	0.223	0.051
(pg/mL)	Post	15.41 ± 3.79	9.95 ± 6.31			
TNF-a	Pre	28.21 ± 8.51	27.28 ± 12.35	0.039	0.555	0.001
(pg/mL)	Post	36.68 ± 11.68	20.77 ± 8.64			
CRP	Pre	34.33 ± 15.80	33.19 ± 10.30	0.001	0.013	0.001
(pg/mL)	Post	54.65 ± 11.66	26.65 ± 8.13			
RSTN	Pre	5.92 ± 1.98	5.56 ± 2.55	0.001	0.859	0.005
(ng/mL)	Post	8.43 ± 4.06	3.33 ± 1.09			
CEA	Pre	2.12 ± 1.21	2.24 ± 0.66	0.034	0.864	0.001
(ng/mL)	Post	2.88 ± 1.07	1.42 ± 0.19			
CK	Pre	222.92 ± 67.13	218.38 ± 51.82	0.869	0.530	0.761
(IU/L)	Post	230.33 ± 80.64	239.69 ± 65.85			
HDL-C	Pre	50.92 ± 9.82	47.08 ± 9.99	0.819	0.992	0.008
(mg/dL)	Post	46.33 ± 8.63	51.69 ± 7.70			
LDL-C	Pre	134.17 ± 43.47	145.31 ± 35.44	0.332	0.005	0.009
(mg/dL)	Post	131.83 ± 36.60	97.54 ± 23.88			

All data represents the mean ± standard deviation. IL-6, TNF, CRP, RSTN, CEA, CK, HDL-C, and LDL-C mean interleukin-6, tumor necrosis factor, C-reactive protein, resistin, carcinoembryonic antigen, creatine kinase, high-density lipoprotein-cholesterol, and low-density lipoprotein-cholesterol, respectively.

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
