# Peer review of "EMS-effect of Exercises with Music on Fatness and Biomarkers of Obese Elderly Women"

_medicina, 2020, doi:10.3390/medicina56040158_

Round 1

Reviewer 1 Report

  1. The work is of great interest, both from the theoretical and especially from the practical side. Since the problem of obesity has become a big social problem, on the one hand. On the other hand. the use of electromyostimulation to maintain health, human performance and rehabilitation of patients suffering from sarcopenia, due to old age, have the most important medical and social significance, being the most important component of strengthening the health of the nation. In the latter case, it is fundamentally important if, as shown by the authors, an increase in skeletal muscle mass is noted.
  2. Overall an interesting look at the physiological consequences of electromyostimulation and the manuscript is of great interest in its field of research
  3. The authors assessed the effect of electromyostimulation on the changes in body composition and cytokines during the course of providing elderly people with aerobic dance. I read carefully this manuscript and found several serious concerns in your study. First of all, there was no control group - groups of subjects using only the electrostimulation procedure. This would be needed to compare groups.
  4. I found some serious concerns. First, authors mainly described the changes body composition and cytokines but you didn’t measure the change in their overall working capacity which is extremely important for an elderly person.
  5. The hypothesis of research is not presented. Author should do so
  6. Line 26-27 Indicated that the results "The results indicate that a 26 significant effect due to the EMS intervention was found concerning bod...." but Line 374-375 Indicated that "Since most of the biochemical variables did not show positive changes in most of the results of this study..."
  7. Line 192 Describe “   EMS suit..”. or specify the link. Need refs here
  8. The mass of muscles of the upper and lower extremities varies significantly and different current amplitudes are required to excite them. Did you take this into account in your research? Author should do so
  9. The work does not indicate the procedure of electrical stimulation itself: the position of the electrodes and their size, pulse parameters and stimulus regimes... Should mention for us
  10. How did the stimulation take place: all muscles simultaneously? Author should do so
  11. Stimulation was applied at what point in an arbitrary movement? Author should do so

Author Response

Answers to reviewer’s comments 

Thank you for your kind advice and comments for publication in Medicina. We revised the manuscript as per your comments. We represented the specific modifications in response to the comments by blue-letters in our manuscript. We sincerely appreciate your comments because your comments make our manuscript better.

Reviewer 1:

#1. Comments and Suggestions: The work is of great interest, both from the theoretical and especially from the practical side. Since the problem of obesity has become a big social problem, on the one hand. On the other hand, the use of electromyostimulation to maintain health, human performance and rehabilitation of patients suffering from sarcopenia, due to old age, have the most important medical and social significance, being the most important component of strengthening the health of the nation. In the latter case, it is fundamentally important if, as shown by the authors, an increase in skeletal muscle mass is noted.

#1. Response: Thank you for what the reviewer has pointed out above comments. In view of your comments, we inserted sentence “ …serious health complications, such as sarcopenia and loss of lean mass. … In women, obesity aggravates physical conditions [3], which are influenced by negative perceptions about loss of lean mass and body shape [2,4].” on Line 42-45.

#2. Comments and Suggestions: Overall an interesting look at the physiological consequences of electromyostimulation and the manuscript is of great interest in its field of research.

#2. Response: Thank you for what the reviewer has interested in our manuscript. We will try to lead a lot of research in the future.

#3. Comments and Suggestions: The authors assessed the effect of electromyostimulation on the changes in body composition and cytokines during the course of providing elderly people with aerobic dance. I read carefully this manuscript and found several serious concerns in your study. First of all, there was no control group - groups of subjects using only the electrostimulation procedure. This would be needed to compare groups.

#3. Response: Thank you for what the reviewer has pointed out above comments. This paper attempted to determine if providing electrical stimulation during aerobic dance would help improve obesity and blood inflammatory factors in elderly women. In other words, the group presented as a control group in this study refers to a group that is not electrically stimulated. However, it seems to be a bit confusing when readers read the paper. So I corrected the Line 104 to Line 108 sentences.

#4. Comments and Suggestions: I found some serious concerns. First, authors mainly described the changes body composition and cytokines but you didn’t measure the change in their overall working capacity which is extremely important for an elderly person.

#4. Response: Thank you for what the reviewer has pointed out above comments. In fact, at the beginning of our paper, we examined both the daily activity and the meals of the elderly you pointed out. However, the volume of the paper is beyond the limit and this part is not included. We attached this part by your point of view as below:

2.5. Calorie intake / output, daily physical activity, and working capacity

This study investigated the calorie intake / output and daily physical activity in order to control and minimize the extrinsic variables that may affect the results of the experiment. Above all, the working capacity or maximum oxygen uptake was measured by exercise test to know the limit of physical activity of elderly women. Prior to the experiment, the obese patients were provided a diary to record what they consumed for a day throughout the experimental period. During this time, we calculated their daily caloric intake volume using CAN-Pro 5.0 (Korean Nutrition Society, Korea) every day for 8 weeks. The daily amount of physical activity that was performed outside the experiment was recorded and calculated. The patients answered the questionnaires based on the recordings of physical activity for the past 7 days for 8 weeks. The total score was obtained through the summation of the duration (in minutes) and frequency (days) of walking, moderate-intensity activity, and vigorous-intensity activity. Then, the data were used to calculate the amount of physical activity based on the international physical activity score conversion method using the metabolic equivalent (MET)-minutes score as shown in Table 2 [19]. Finally, the calorie intake, calorie output, and physical activity were recorded and the mean values of the above variables for 4 weeks were analyzed.

Table 2. Degrees of category scores by international physical activity questionnaire

Category

Criteria

#

Degree

1

Low

activity

• No activity is reported OR

• Some activity is reported but not enough to meet Categories 2 or 3.

2

Moderate

activity

Either of the following 3 criteria

• 3 or more days of vigorous activity of at least 20 minutes per day OR

• 5 or more days of moderate-intensity activity and/or walking of at least 30 minutes per day OR

• 5 or more days of any combination of walking, moderate-intensity or vigorous-intensity.

3

High

activity

Any one of the following 2 criteria

• Vigorous-intensity activity on at least 3 days and accumulating at least 1,500 MET-minutes/week OR

• 7 or more days of any combination of walking, moderate- or vigorous-intensity activities accumulating at least 3,000 MET-minutes/week.

Equations for calculating physical activity degree as follows; Walking MET-min/week = 3.3 × min of activity/day × days per week. Moderate-intensity physical activity MET-min/week = 4.0 × min of activity/day × days per week. Vigorous-intensity physical activity MET-min/week = 8.0 × min of activity/day × days per week. Total MET-minutes/week = Walking MET-min/week + Moderate-intensity physical activity MET-min/week + Vigorous-intensity physical activity MET-min/week.

Maximum oxygen intake was measured only once before the experiment to determine the degree of difficulty during aerobic dancing. A gas analyzer (Qurak CPET®, Cosmed, Italy), an ECG analyzer (Heartwave II®, Cambridge Heart Inc., USA) and a treadmill ergometer (T150, HP/Cosmos®, Germany) were used for this experiment. For investigating the working capacity of elderly women, the modified Bruce protocol was inputted for the graded exercise testing and the speed and gradient controls were checked on the treadmill. All patients were restrained from vigorous physical activity and taking medication 48 hours prior to the test and eating 3 hours prior to the test. The electrodes were attached to the chest and the blood pressure cuff was place on the brachial artery. Mouth piece was fixed over the lip and nose area to breath only through the mouth and to the mouth piece. The test was conducted after pre-checking the electrocardiography, automatic sphygmomanometry, and sphygmometer for proper operation. The patients were encouraged to reach the all-out state according to modified Bruce protocol [63]. The treadmill was halted and the subjects were seated to monitor the recovery cardiopulmonary variables after the test.

  1. Jee, Y.S. The effect of high-impulse electromyostimulation on adipokine profiles, body composition and strength: A pilot study. Isokin. Exerc. Sci. 2019, 27, 163–176. doi 10.3233/IES-183201.
  2. Kim, S.E.; Hong, J.; Cha, J.Y.; Park, J.M.; Eun, D.; Yoo, J.; Jee, Y.S. Relative appendicular skeletal muscle mass is associated with isokinetic muscle strength and balance in healthy collegiate men. J Sports Sci. 2016, 34, 2114-2120. doi: 10.1080/02640414.2016.1151922.

Also, we attached the results of this part by your point of view as a part of Results:

“3.1. Comparison of demographics, calorie intake/output, physical activity, and working capacity

There were no significant differences between groups for all variables as shown in Table 1. The maximum oxygen uptakes of CON and EMSG were 23.13 ± 4.25 ㎖/kg/ min and 25.34 ± 6.51 ㎖/kg/min, respectively, and there was no significant difference between groups. In particular, percent fat was not significantly different between groups. There were also no significant differences between groups in mean values of calorie intake, calorie output, and physical activity level during the experimental period as shown in Table 3.

Table 3 Differences of controlled variables

Items

Week

Groups

CON (n = 12)

EMSG (n = 13)

Z

p

Calorie intake

(kcal)

4

1663.21 ± 124.38

1671.36 ± 182.25

-0.842

0.369

8

1688.69 ± 119.85

1641.43 ± 193.62

-0.279

0.823

Calorie output

(kcal)

4

246.25 ± 51.23

258.08 ± 49.25

-0.771

0.482

8

229.97 ± 52.36

237.88 ± 51.69

-0.256

0.897

PAC

scores

4

1.85 ± 1.45

1.83 ± 1.47

-0.129

0.989

8

1.84 ± 1.63

1.87 ± 1.78

-0.131

0.971

All values are expressed as mean ± standard deviation. CON, EMSG, and PAC represent control group, electromyostimulation group, and physical activity category, which scored by 1 (low), 2 (moderate), and 3 (high) activity levels.

#5. Comments and Suggestions: The hypothesis of research is not presented. Author should do so.

#5. Response: Thank you for what the reviewer has pointed out above comments. In view of your comments, I inserted the hypothesis in the end of introduction as follow: “The hypothesis of this study was divided into two as follows. First, the combination of aerobic dance and EMS would bring about positive changes in body composition. Second, a combination of aerobic dance and EMS would lead to positive changes in cytokines”.

#6. Comments and Suggestions: Line 26-27 Indicated that the results "The results indicate that a 26 significant effect due to the EMS intervention was found concerning bod...." but Line 374-375 Indicated that "Since most of the biochemical variables did not show positive changes in most of the results of this study...“

#6. Response: Thank you for what the reviewer has pointed out above comments. The meaning of this sentence was that CRP and HDL-C did not change significantly in the EMSG population. However, this sentence would be likely to confuse the reader and modified it as follows: “However, since CRP and HDL-C of the biochemical variables did not show positive changes in the EMSG,…”

#7. Comments and Suggestions: Line 192 Describe “ EMS suit..”. or specify the link. Need refs here.

#7. Response: Thank you for what the reviewer has pointed out above comments. According to your comments, I inserted reference “…EMS suit [19]”.

#8. Comments and Suggestions: The mass of muscles of the upper and lower extremities varies significantly and different current amplitudes are required to excite them. Did you take this into account in your research? Author should do so.

#8. Response: Thank you for what the reviewer has pointed out above comments. Of course, as the reviewer pointed out, 8 parts of the upper and lower body were measured and managed by Bluetooth as shown below. For avoiding to confuse, the below sentence was inserted as follows: “Each 1 MT of the upper and lower body was measured and stored in Bluetooth, and the intensity was adjusted for each individual during aerobic dancing.”

#9. Comments and Suggestions: The work does not indicate the procedure of electrical stimulation itself: the position of the electrodes and their size, pulse parameters and stimulus regimes... Should mention for us.

#9. Response: Thank you for what the reviewer has pointed out above comments. For avoiding to confuse, the below sentence was inserted as follows: “Participants were given variously sized EMS suits made by Miracle®(Seoul, Korea) according to their size. The suits were composed of a silicone conductive pad and wireless materials. The electrical strength of the suit was controlled via Bluetooth. EMS suits used in this study enabled the simultaneous activation of eight muscle groups (both upper legs, both upper arms, buttocks, abdomen, chest, lower back, upper back, and latissimus dorsi) with selectable intensities for each region. Based on recommendations from available literature [9,20,25-27], the stimulation frequency was selected at 85 Hz, the impulse-width at 350 μs, the impulse-rise as a rectangular application…”

#10. Comments and Suggestions: How did the stimulation take place: all muscles simultaneously? Author should do so.

#10. Response: Thank you for what the reviewer has pointed out above comments. As explained above, I inserted the sentence on Line 226-228 as follow: “Each 1 MT of the upper and lower body was measured and stored in Bluetooth, and the intensity was adjusted for each individual during aerobic dancing.”.

#11. Comments and Suggestions: Stimulation was applied at what point in an arbitrary movement? Author should do so.

#11. Response: Thank you for what the reviewer has pointed out above comments. The impulse duration of stimulation was 6 sec during aerobic dancing, with a 4-sec break between impulses. I inserted the sentence on Line 237-238 as follow: “…the impulse duration was 6 s, with a 4-s break between impulses…”

We’ve got the English Editing Service through https://www.mdpi.com/authors/english.

Thank you for your comments, we represented the modifications in response to your comments.

March 7, 2020

Reviewer 2 Report

Manuscript ID: medicina-735529 Aerobic Dance Wore with EMS Suit Improves Fatness and Biomarkers of Obese Elderly Women 

The authors write that “The elderly population is increasing all over the world, and the incidence rate of chronic degenerative diseases is also increasing rapidly.” I deeply agree with it. And looking for new methods to help elderly people to cope with that situation and encouraging them to take part in a different kind of exercise/training is necessary. That is why I find this article to be very interesting for all of us, not only for researchers working on the effect of an effort on people's health.

Comments:

Title:

Expression “Aerobic dance” is unclear because it suggests that there is also anaerobic dance which is rather impossible. That is why I recommend using just “aerobics” or “exercises with music”. Please, consider this in entire body text (e.g. line 96, 104, 177, 199, etc.).

Materials and Methods section: 

Line 104: Authors write that “All patients knew…”. It is unclear. Does it mean both groups? 

Table 1:

  • In my opinion, the sentences below the table “All data represent the mean ± standard deviation. EMS and CON mean the electromyostimulation group and control group, respectively. Symbol * was analyzed by the Mann–Whitney U test.” are should be placed in the “Data Analysis” section.
  • If the “Z” symbol describes the difference between CON data and EMSG data, why the authors did not use “D” which is commonly used for “difference”.
  • The authors write “Symbol * was analyzed by the Mann–Whitney U test.” which is unclear because it suggests that the star was analyzed, not “difference”. Please, correct this.

Lines 196 – 198:

The authors write “… the dance movements were simplified and composed of clapping and tapping, bending and rotating, aerobic and anaerobic exercises…” although they placed “aerobic dance” in the title of the paper and in other parts of the text. Please, explain this because it can be understandable for readers.

Line 215

There is “.05”, and it should be “0.05”. Am I right?

Line 216

There is “.95”, and it should be “0.95”

Results section:

Line 227

The authors write “The percent of fat of CON increased, although not significantly…”. In fact, there was no change. I recommend not using that kind of expression when there is no significant change.

Tables 2 and 3

Comments to the sentences under the tables are like those of Table 1.

In my opinion, the "star" symbol is incorrectly used. It is commonly known that one star is used for p<0.05 and two stars for p<0.001. Such use of these symbols may be incomprehensible to readers.

Figure 1.

In my opinion, the figure is useless because it duplicates the data from the table. Besides, it is not clear why the authors decided to present only one parameter graphically, i.e. fat mass, and no other parameters. Besides, the statistical significance can be shown in the Table. The question also arises whether the star symbols mean the same in the tables and figures.

Figure 2.

The same comments as in figure 1’s case.

Figure 3.

The same comments as in figure 1’s case. 

Discussion section:

The authors write that “… almost all biomarkers in CON showed no change…” which is not a precise statement. Please, write which ones.

Other comments:

I think it will be very useful if the authors place the “Limitation of experiment” section.

Author Response

Answers to reviewer’s comments 

Thank you for your kind advice and comments for publication in Medicina. We revised the manuscript as per your comments. We represented the specific modifications in response to the comments by blue-letters in our manuscript. We sincerely appreciate your comments because your comments make our manuscript better.

Reviewer 2:

The authors write that “The elderly population is increasing all over the world, and the incidence rate of chronic degenerative diseases is also increasing rapidly.” I deeply agree with it. And looking for new methods to help elderly people to cope with that situation and encouraging them to take part in a different kind of exercise/training is necessary. That is why I find this article to be very interesting for all of us, not only for researchers working on the effect of an effort on people's health.

#1. Comments and Suggestions about the Title:

Expression “Aerobic dance” is unclear because it suggests that there is also anaerobic dance which is rather impossible. That is why I recommend using just “aerobics” or “exercises with music”. Please, consider this in entire body text (e.g. line 96, 104, 177, 199, etc.).

#1. Response: Thank you for what the reviewer has pointed out above comments. In view of your comments, we selected the exercises with music ~.

#2. Comments and Suggestions about the Materials and Methods section:

Line 104: Authors write that “All patients knew…”. It is unclear. Does it mean both groups?

#2. Response: Thank you for what the reviewer has pointed out above comments. Yes, here ‘all patients’ mean both group.

#3. Comments and Suggestions about the Materials and Methods section:

Table 1:

In my opinion, the sentences below the table “All data represent the mean ± standard deviation. EMS and CON mean the electromyostimulation group and control group, respectively. Symbol * was analyzed by the Mann–Whitney U test.” are should be placed in the “Data Analysis” section.

If the “Z” symbol describes the difference between CON data and EMSG data, why the authors did not use “D” which is commonly used for “difference”. The authors write “Symbol * was analyzed by the Mann–Whitney U test.” which is unclear because it suggests that the star was analyzed, not “difference”. Please, correct this.

#3. Response: Thank you for what the reviewer has pointed out above comments. According to your comments, we erased or corrected a word below Table 1.

#4. Comments and Suggestions about the Materials and Methods section:

Lines 196 – 198:

The authors write “… the dance movements were simplified and composed of clapping and tapping, bending and rotating, aerobic and anaerobic exercises…” although they placed “aerobic dance” in the title of the paper and in other parts of the text. Please, explain this because it can be understandable for readers.

#4. Response: Thank you for what the reviewer has pointed out above comments. In view of your comments, we changed above sentences as follow: “ In order to provide effective muscular contractions and to prevent harmful joint injury, the dance movements were simplified and composed of clapping and tapping, bending and rotating, exercises with music, and stretching exercises, which were performed according to the instructor's directions.”

#5. Comments and Suggestions about the Materials and Methods section:

Line 215

There is “.05”, and it should be “0.05”. Am I right?

#5. Response: Thank you for what the reviewer has pointed out above comments. In view of your comments, we changed as follow: size of f²(V) = 0.35 (medium size effect), α error probability of 0.05, and power (1 − β error probability) of 0.95.

#6. Comments and Suggestions about the Materials and Methods section:

Line 216

There is “.95”, and it should be “0.95”

#6. Response: Thank you for what the reviewer has pointed out above comments. In your comments, ‘0’ is all righted. So, I inserted ‘0’ in all sentences.

#7. Comments and Suggestions about the Results section:

Line 227

The authors write “The percent of fat of CON increased, although not significantly…”. In fact, there was no change. I recommend not using that kind of expression when there is no significant change.

#7. Response: Thank you for what the reviewer has pointed out above comments. In view of your comments, we deleted all kind of expression when there is no significant change as follows: “As shown in Table 3, the body weight of CON did not change, while that of EMSG decreased significantly. These results showed differences between groups after 8 weeks. The skeletal muscle mass of EMSG increased significantly (p = 0.009), whereas that of CON did not change, indicating a significant difference between groups (p = 0.019). Although the fat mass of EMSG was significantly decreased while that of CON was not changed, there was a significant difference between groups. The percent of fat of CON increased, while that of EMSG showed a tendency to decrease. This result showed a significant difference between groups after the experiment. Specifically, the basal metabolic rate (BMR) of EMSG significantly increased, whereas that of CON significantly decreased, by the end of the experiment.”

#8. Comments and Suggestions about the Results section:

Tables 2 and 3

Comments to the sentences under the tables are like those of Table 1.

In my opinion, the "star" symbol is incorrectly used. It is commonly known that one star is used for p<0.05 and two stars for p<0.001. Such use of these symbols may be incomprehensible to readers.

#8. Response: Thank you for what the reviewer has pointed out above comments. In view of your comments, we changed all things below the Tables.

#9. Comments and Suggestions about the Results section:

Figure 1, Figure 2, Figure 3

In my opinion, the figure is useless because it duplicates the data from the table. Besides, it is not clear why the authors decided to present only one parameter graphically, i.e. fat mass, and no other parameters. Besides, the statistical significance can be shown in the Table. The question also arises whether the star symbols mean the same in the tables and figures.

#9. Response: Thank you for what the reviewer has pointed out above comments. According view of your comments, we deleted all Figure.

#10. Comments and Suggestions about the Discussion section:

The authors write that “… almost all biomarkers in CON showed no change…” which is not a precise statement. Please, write which ones.

#10. Response: Thank you for what the reviewer has pointed out above comments. In view of your comments, we changed as follows: “This study found some evidence that exercises with music wearing EMS suits improved body composition. Body weight and fat mass in EMSG were significantly changed. Additionally, almost all biomarkers in CON showed no change after 8 weeks, although the cytockines significantly decreased in EMSG. Specifically, LDL-C in EMSG was significantly decreased by the end of the experiment. These results revealed significant differences between the groups after 8 weeks.”

#11. Comments and Suggestions about the Others:

I think it will be very useful if the authors place the “Limitation of experiment” section.

#11. Response: Thank you for what the reviewer has pointed out above comments. In view of your comments, we inserted the Limitation in conclusion as follows: “Although this study showed positive results in body composition and biomarkers, the sample size and larger trials are strongly needed to obtain excellent results.”

We’ve got the English Editing Service through https://www.mdpi.com/authors/english.

Thank you for your comments, we represented the modifications in response to your comments.

March 7, 2020

Reviewer 3 Report

Dear authors,

thank you for submitting your manuscript to medicina.

In the following, you can find some remarks regarding the correction and some questions that has to be answered before the possibility to publish this article. In general, you talk about EMS (what is mostly understood as locally stimulation) and totally miss the whole-body EMS (what it actually is). For better understanding, a more precise definition should be given:

Abstract:

  • l.16: not only eight muscles were stimulated, it was eight pairs of electrodes

Introduction:

  • l.48: If it is not easy for elderly to participate in exercises, why did you choose aerobic dancing?
  • l.64: missing literature for the loose of body fat because of EMS

Materials and Methods:

  • l.96: Where did you search for participants? How many wanted to join the study, how many drop outs after allocation did you have? I would prefer a flow of participants for a better understanding
  • l.120: Why did you choose the Mann-Whitney U test?
  • l 130: You used two BIA to measure the body composition. Did you get the mean value of both or why did you choose two measuring systems?
  • l.138: Was there any controle of the water intake of the participants to avoid deviations in the BIA measurements?
  • l.192: The participants trained with their individual percentage of MT depending on the week of training. Why did you specify the RPE (l.199) if the intensity was already specified via % of MT? Did CON also train with the RPE? Which RPE scale was used (literature on this is missing)?
  • l.179 + 195 + 201: If you followed existing guidelines for EMS, why did you train 3 times a week for 40 minutes? This exceeds all current safety requirements (Kemmler et al. 2018, 2019), furthermore no physiological control was performed (e.g. in the form of creatine kinase). How did you ensure that you did not completely overload the subjects? Current results would indicate exactly this with such a large amount of training and would consider such a procedure, especially with such a sample, to be very critical.
  • l.214: What was the determined sample size? You said how you calculated it, but not the results. It is unclear how the sample size could be that small, especially when recalculating the sample size with your values in G*Power. A G*Power Output for a better understanding would be good.
  • l.216 ff: Why didn´t you choose a multifactorial ANOVA? Even if there is no normal distribution, the ANOVA is robust against it and therefore the procedure to be better applied. Furthermore, the alpha error accumulation is not taken into account, even if it is necessary because of the multiple testing. I doubt that the results are exactly the same after the correction. Please provide the results of the ANOVA or recalculate the values with the alpha error correction

Discussion:

  • l.287 ff: In the study of Anderson et al., 5x30 minute walking per week improved markers of the CON group without EMS. They made a comparison of the groups using the measured mean RPE values of the exercises (walking RPE mean: 12,3 ; walking+EMS RPE mean: 12,4 --> similar to the present study). Would this be a useful approach to compare your groups?
  • Also further studies showed positive effects of light aerobic dance or other aerobic procedures on elderly women (Hopkins 1995; Arslan 2011; Mc Cord et al. 1989...). It is totally unclear, why those results could not be confirmed in your study. The discussion and especially the limitation section is too small. If the reason for the missing improvements in CON really is because of the diet, you have to highlight this fact and try to explain, why the diet influences all markers and even make some of them worse in post than in pre. Furthermore, the saple size should be highlighted as a limitation

In particular, the statistics and the lack of improvement of the CON need to be elaborated in more detail. There must be a reason for the lack of improvement, because a training of 3x40 minutes over 8 weeks with an RPE of 11 on average with the trained test person definitely indicates improvements.

Kind Regards

Author Response

Answers to reviewer’s comments 

Thank you for your kind advice and comments for publication in Medicina. We revised the manuscript as per your comments. We represented the specific modifications in response to the comments by blue-letters in our manuscript. We sincerely appreciate your comments because your comments make our manuscript better.

Reviewer 3:

In the following, you can find some remarks regarding the correction and some questions that has to be answered before the possibility to publish this article. In general, you talk about EMS (what is mostly understood as locally stimulation) and totally miss the whole-body EMS (what it actually is). For better understanding, a more precise definition should be given:

#1. Comments and Suggestions about the Abstract:

l.16:

not only eight muscles were stimulated, it was eight pairs of electrodes

#1. Response: Thank you for what the reviewer has pointed out above comments. In view of your comments, we changed from muscles to pairs...

#2. Comments and Suggestions about the Introduction:

l.48:

If it is not easy for elderly to participate in exercises, why did you choose aerobic dancing?

#2. Response: Thank you for what the reviewer has pointed out above comments. The criterion to select subjects in this study was to randomly select the elderly who applied to enter exercises with music. For avoiding a confused sentence, I inserted the sentence as follows: “…and then selected subjects were to randomly classified the elderly who applied to enter exercises with music.”

#3. Comments and Suggestions about the Introduction:

l.64:

missing literature for the loose of body fat because of EMS

#3. Response: Thank you for what the reviewer has pointed out above comments. In view of your comments, we inserted reference.

#4. Comments and Suggestions about the Materials and Methods:

l.96:

Where did you search for participants? How many wanted to join the study, how many drop outs after allocation did you have? I would prefer a flow of participants for a better understanding

#4. Response: Thank you for what the reviewer has pointed out above comments. In view of your comments, we inserted Fig.1 and then explained about the number of drop-out as follows: “After excluding two patients out of thirty-two eligible participants, the remaining thirty patients belonged to one of two groups. Of the 15 patients in the CON who were allocated to the non-EMS group, one did not receive assessment and two were lost in the follow-up phase. Therefore, 12 patients in the CON were analyzed in our study. Furthermore, of the 15 patients in the EMSG, one did not receive assessment and another was lost in the follow-up phase. Therefore, 13 patients of the EMSG were analyzed in our study as shown in Fig. 1.”

Fig. 1. Patients’ allocation (consolidated standards for reporting of trials flow diagram).

#5. Comments and Suggestions about the Materials and Methods:

l.120:

Why did you choose the Mann-Whitney U test?

#5. Response: Thank you for what the reviewer has pointed out above comments. In our opinion, based on the results of the Shapiro–Wilk test, the normality of groups was not distributed. Therefore, we inserted the sentences as follows: “All data are reported as the mean (SD). The sample size was determined using G*Power v 3.1.3, considering an a priori effect size f²(V) = 0.35 (medium size effect), α error probability = 0.05 and power (1 − β error probability) = 0.95. A sample size of 30 was recommended, and the current sample included 25 participants. Based on the results of the Shapiro–Wilk test, we analyzed the data using non-parametric tests. Mann–Whitney U test and Wilcoxon rank test were used to examine the differences of variables between groups and to examine the changes of variables between times. A significance of p < 0.05 was employed. SPSS 18.0 (SPSS Inc., Chicago, IL, USA) was used for all analyses.”

#6. Comments and Suggestions about the Materials and Methods:

l.130:

You used two BIA to measure the body composition. Did you get the mean value of both or why did you choose two measuring systems?

#6. Response: Thank you for what the reviewer has pointed out above comments. For avoiding confused sentence, we inserted the sentence as follows: “To measure body composition, all patients were weighed while wearing light clothes and no shoes. The bioelectrical impedance analysis was employed using the BMS 330 for height and InBody 320 for body composition (Biospace Co., Ltd., Korea), respectively.”

#7. Comments and Suggestions about the Materials and Methods:

l.138: 

Was there any control of the water intake of the participants to avoid deviations in the BIA measurements?

#7. Response: Yes. this is the usual way to take before applying the BIA (bioelectrical impedance analysis) method.

#8. Comments and Suggestions about the Materials and Methods:

l.192:

The participants trained with their individual percentage of MT depending on the week of training. Why did you specify the RPE (l.199) if the intensity was already specified via % of MT? Did CON also train with the RPE? Which RPE scale was used (literature on this is missing)?

#8. Response: The reason why RPE was measured each time was to measure the difficulty of subjects during EMS combined exercises with a music. This was measured in case the subjects could not digest the combined program course, and was a preliminary measure to accept if they wanted to drop out of the course.

#9. Comments and Suggestions about the Materials and Methods:

l.179 + 195 + 201:

If you followed existing guidelines for EMS, why did you train 3 times a week for 40 minutes? This exceeds all current safety requirements (Kemmler et al. 2018, 2019), furthermore no physiological control was performed (e.g. in the form of creatine kinase). How did you ensure that you did not completely overload the subjects? Current results would indicate exactly this with such a large amount of training and would consider such a procedure, especially with such a sample, to be very critical.

#9. Response: Your opinion is also correct. However, considering the disagreement among authors, we have completed the safety test of EMS and conducted this study based on the results. After the test, the creatine kinase level in the elderly was normal. The reference is shown in 17 below.

  1. Jee, Y.S. The efficacy and safety of whole-body electromyostimulation in applying to human body: based from graded exercise test. J. Exerc. Rehabil. 2018, 14, 49-57. doi: 10.12965/jer.1836022.011.

#10. Comments and Suggestions about the Materials and Methods:

l.214:

What was the determined sample size? You said how you calculated it, but not the results. It is unclear how the sample size could be that small, especially when recalculating the sample size with your values in G*Power. A G*Power Output for a better understanding would be good.

#10. Response: As mentioned above, the sample size was determined using G*Power v 3.1.3. A sample size of 30 was recommended, and the current sample included 25 participants. Based on the results of the Shapiro–Wilk test, we analyzed the data using non-parametric tests. Mann–Whitney U test and Wilcoxon rank test were used to examine the differences of variables between groups and to examine the changes of variables between times.

#11. Comments and Suggestions about the Materials and Methods:

l.216 ff:

Why didn´t you choose a multifactorial ANOVA? Even if there is no normal distribution, the ANOVA is robust against it and therefore the procedure to be better applied. Furthermore, the alpha error accumulation is not taken into account, even if it is necessary because of the multiple testing. I doubt that the results are exactly the same after the correction. Please provide the results of the ANOVA or recalculate the values with the alpha error correction

#11. Response: Like you said it was a good way too. However, the data was not significant in the normality distribution, so we chose nonparametric analysis.

#12. Comments and Suggestions about the Discussion:

l.287 ff:

In the study of Anderson et al., 5x30 minute walking per week improved markers of the CON group without EMS. They made a comparison of the groups using the measured mean RPE values of the exercises (walking RPE mean: 12,3 ; walking+EMS RPE mean: 12,4 --> similar to the present study). Would this be a useful approach to compare your groups?

Also further studies showed positive effects of light aerobic dance or other aerobic procedures on elderly women (Hopkins 1995; Arslan 2011; Mc Cord et al. 1989...). It is totally unclear, why those results could not be confirmed in your study. The discussion and especially the limitation section is too small. If the reason for the missing improvements in CON really is because of the diet, you have to highlight this fact and try to explain, why the diet influences all markers and even make some of them worse in post than in pre. Furthermore, the saple size should be highlighted as a limitation. In particular, the statistics and the lack of improvement of the CON need to be elaborated in more detail. There must be a reason for the lack of improvement, because a training of 3x40 minutes over 8 weeks with an RPE of 11 on average with the trained test person definitely indicates improvements.

#12. Response: Thank you for what the reviewer has pointed out above comments. This part could not be attached because it exceeded the length of the paper. According to the reviewer's point of view, the contents were attached to the main text as follows, and the discussion section was rewritten.

 “3.1. Comparison of demographics, calorie intake/output, physical activity, and working capacity

There were no significant differences between groups for all variables as shown in Table 1. The maximum oxygen uptakes of CON and EMSG were 23.13 ± 4.25 ㎖/kg/ min and 25.34 ± 6.51 ㎖/kg/min, respectively, and there was no significant difference between groups. In particular, percent fat was not significantly different between groups. There were also no significant differences between groups in mean values of calorie intake, calorie output, and physical activity level during the experimental period as shown in Table 3.

Table 3 Differences of controlled variables

Items

Week

Groups

CON (n = 12)

EMSG (n = 13)

Z

p

Calorie intake

(kcal)

4

1663.21 ± 124.38

1671.36 ± 182.25

-0.842

0.369

8

1688.69 ± 119.85

1641.43 ± 193.62

-0.279

0.823

Calorie output

(kcal)

4

246.25 ± 51.23

258.08 ± 49.25

-0.771

0.482

8

229.97 ± 52.36

237.88 ± 51.69

-0.256

0.897

PAC

scores

4

1.85 ± 1.45

1.83 ± 1.47

-0.129

0.989

8

1.84 ± 1.63

1.87 ± 1.78

-0.131

0.971

All values are expressed as mean ± standard deviation. CON, EMSG, and PAC represent control group, electromyostimulation group, and physical activity category, which scored by 1 (low), 2 (moderate), and 3 (high) activity levels.

In the aspect of Discussion, a part of sentences was changed as follows: “Meanwhile, the reason that there was negative or no change in the variables of CON is thought to be due to a small range of dancing activities for avoiding the severe change of degenerative joints. Also, only 8 weeks of exercises with music in the elderly obese women were short, suggesting no changes in body composition or cytokines.” On Line 346-347.

We’ve got the English Editing Service through https://www.mdpi.com/authors/english.

Thank you for your comments, we represented the modifications in response to your comments.

March 7, 2020

Round 2

Reviewer 1 Report

  1. The use of electrical stimulation training in a variant, as the author describes, does not allow attributing the results to the positive influence of electrical training. Possibly it is, but where the results of the impact of only such a training. Author should do so.
  2. Page 4, line 182 The author indicates that the maximum oxygen absorption was measured using a test with physical activity, but does not describe the determination procedure, does not indicate the power to fulfill the load, an indicator indicating the achievement of a peak in oxygen consumption, and so on. The method of determination is not specified, i.e. direct method or calculation method, although the author indicates that the working capacity or maximum oxygen uptake was measured using a physical exercise test. If the calculated author must indicate the accuracy of the predicted method.
  3. Page 4, line 185 Performance or maximum oxygen uptake. The maximum oxygen uptake is not operability, but an indicator by which you can evaluate the level of working capacity. Author should do so.
  4. Page 6, line 206 Criterion or indicator that would confirm the aerobic nature of the exercises performed. Author should do so.
  5. Page 6, line 219 Music Program What it is?
  6. Page 7, line 220 Further, the author lists a number of exercises but again does not indicate a criterion for assessing the intensity of the exercises performed, and especially information on stretching exercises. Author should do so.
  7. Page 6, line 230-231 Each 1 MT ...and stored ... and the intensity was adjusted for each individual during aerobic dancing. The intensity of the electrical workout was different from maximal tolerance (1 MT) Author should do so.
  8. How coordination of movements has changed with a combination of voluntary movement plus the evoked contraction during exercise. Have biomechanical motion parameters been recorded? Author should do so.
  9. And most importantly. If we assume that the results obtained are the influence exerted by an electric training, then what is the mechanism of influence of such a training? Author should do so.

Author Response

Answers to reviewer’s comments 

Thank you for your kind advice and comments for publication in Medicina. We re-revised the manuscript as per your comments. We represented the specific modifications in response to the comments by blue-letters in our manuscript. We sincerely appreciate your comments because your comments make our manuscript better.

Reviewer 1:

#1. Comments and Suggestions:

The use of electrical stimulation training in a variant, as the author describes, does not allow attributing the results to the positive influence of electrical training. Possibly it is, but where the results of the impact of only such a training. Author should do so.

#1. Response: Thank you for what the reviewer has pointed out above comments. According to the instructions of other reviewers, all statistical methods were changed to ANOVA, and when looking at the p value of interaction (G * T) in the results, significant results were found in body composition and biomarkers. Sorry for the confusion. Details of these results are as follows:

“  3.2. Effect of EMS on Body Composition

As shown in Table 4, no significant effect of the EMS intervention was found for body weight when comparing the intervention and control groups. However, skeletal muscle mass (F = 7.826), fat mass (F = 8.717), percent fat (F = 4.961), and the basal metabolic rate (BMR) (F = 28.770) were significantly different in terms of the group by time interaction. This result shows a significant effect of the EMS intervention concerning body composition.

Table 4. Differences and changes in body composition

Items

Groups

ANOVA (p)

CON (n = 12)

EMSG (n = 13)

G

T

G×T

Body weight

Pre

63.75 ± 3.82

63.05 ± 8.78

0.517

0.010

0.297

(kg)

Post

62.33 ± 4.17

57.54 ± 6.58

Skeletal muscle

Pre

20.46 ± 0.97

20.94 ± 2.03

0.020

0.285

0.010

(kg)

Post

19.82 ± 1.71

22.41 ± 2.19

Fat mass

Pre

25.23 ± 2.93

26.50 ± 5.50

0.649

0.014

0.007

(kg)

Post

25.42 ± 2.66

22.31 ± 3.89

Percent fat

Pre

38.33 ± 4.31

39.21 ± 1.76

0.273

0.466

0.036

(%)

Post

39.66 ± 2.32

34.95 ± 3.82

Basal Metabolic

Pre

1187.00 ± 35.92

1188.69 ± 67.98

0.004

0.249

0.001

Rate (kcal)

Post

1110.17 ± 40.62

1237.77 ± 74.40

3.3. Effect of EMS on Biomarkers

As shown in Table 5, the level of IL-6 in CON showed an increasing tendency from Week 0 to Week 8, but this level showed a decreasing tendency in EMSG, although there was no significant difference between groups. However, TNF-α (F = 21.003), CRP (F = 27.825), RSTN (F = 9.520), and CEA (F = 19.331) showed significant differences for the group by time interaction. These positive changes were also represented in the HDL-C and LDL-C measurements. Specifically, although the level of CK in both groups showed an increasing tendency after the experiment, no significant difference was found between groups. These results demonstrate significant effects due to the EMS intervention concerning biomarkers in obese, elderly women.

Table 5. Differences and changes in biomarkers

Items (units)

Groups

ANOVA (p)

CON (n = 12)

EMSG (n = 13)

G

T

G×T

IL-6

Pre

14.29 ± 7.52

14.51 ± 7.14

0.236

0.223

0.051

(pg/mL)

Post

15.41 ± 3.79

9.95 ± 6.31

TNF-a

Pre

28.21 ± 8.51

27.28 ± 12.35

0.039

0.555

0.001

(pg/mL)

Post

36.68 ± 11.68

20.77 ± 8.64

CRP

Pre

34.33 ± 15.80

33.19 ± 10.30

0.001

0.013

0.001

(pg/mL)

Post

54.65 ± 11.66

26.65 ± 8.13

RSTN

Pre

5.92 ± 1.98

5.56 ± 2.55

0.001

0.859

0.005

(ng/mL)

Post

8.43 ± 4.06

3.33 ± 1.09

CEA

Pre

2.12 ± 1.21

2.24 ± 0.66

0.034

0.864

0.001

(ng/mL)

Post

2.88 ± 1.07

1.42 ± 0.19

CK

Pre

222.92 ± 67.13

218.38 ± 51.82

0.869

0.530

0.761

(IU/L)

Post

230.33 ± 80.64

239.69 ± 65.85

HDL-C

Pre

50.92 ± 9.82

47.08 ± 9.99

0.819

0.992

0.008

(mg/dL)

Post

46.33 ± 8.63

51.69 ± 7.70

LDL-C

Pre

134.17 ± 43.47

145.31 ± 35.44

0.332

0.005

0.009

(mg/dL)

Post

131.83 ± 36.60

97.54 ± 23.88

on Line 316 to 340 of the new changed manuscript”

#2. Comments and Suggestions: Page 4, line 182 The author indicates that the maximum oxygen absorption was measured using a test with physical activity, but does not describe the determination procedure, does not indicate the power to fulfill the load, an indicator indicating the achievement of a peak in oxygen consumption, and so on. The method of determination is not specified, i.e. direct method or calculation method, although the author indicates that the working capacity or maximum oxygen uptake was measured using a physical exercise test. If the calculated author must indicate the accuracy of the predicted method.

#2. Response: Thank you for what the reviewer has interested in our manuscript. According to the instructions, we inserted the sentences as follows: “In detail, the GXT of this study was used to investigate coronary artery disease and/or abnormal rhythms and to evaluate exercise capacity. The modified Bruce protocol was applied in consideration of the elderly, obese women. In the protocol, the speed stayed constant for the first three stages, starting at 1.7 mph at an incline of 0%. After the third stage, the speed and grade increased by 2.5 mph and 12%, respectively. After that, both the speed and grade increased every 3 min. All subjects were instructed to continue to walk or jog until reaching an all-out level, which was deemed to be their maximal rating of perceived exertion (RPE). During and after walking or running on the treadmill for as long as possible, the subjects were instructed to describe the appearance of any of the following symptoms: chest pain, shortness of breath, dizziness, and leg pain. During the test, subjects were asked to express their level of exercise intensity on the RPE scale. The test was terminated if the following symptoms occurred: (a) a drop in systolic blood pressure of more than 10 mmHg from baseline, despite an increase in workload, when accompanied by other evidence of ischemia; (b) moderate-to-severe angina; (c) an increase in nervous system symptoms; (d) signs of cyanosis; (e) technical difficulties in monitoring electrocardiographic tracings; (f) a subject’s desire to stop; (g) sustained ventricular tachycardia; and (h) ST elevation (>1 mm) in leads without diagnostic Q waves (other than V1 or aVR) [28]. The variable of this study was limited oxygen uptake calculated by body weight every minute at each test stage.” on Line 225 to 242 of the new changed manuscript.

#3. Comments and Suggestions: Page 4, line 185 Performance or maximum oxygen uptake. The maximum oxygen uptake is not operability, but an indicator by which you can evaluate the level of working capacity. Author should do so.

#3. Response: Thank you for what the reviewer has pointed out above comments. According to your suggestion, we inserted the sentence as follows: “The variable of this study was limited oxygen uptake calculated by body weight every minute at each test stage.” On Line 241 to 242 of the new changed manuscript.

#4. Comments and Suggestions: Page 6, line 206 Criterion or indicator that would confirm the aerobic nature of the exercises performed. Author should do so.

#4. Response: Thank you for what the reviewer has pointed out above comments. According to your suggestion, we corrected the sentence as follows: “Maximum oxygen intake was measured only once before the experiment to determine the indicator that would confirm the aerobic nature of the exercises performed during music.” On Line 216 to 217 of the new changed manuscript.

#5. Comments and Suggestions: Page 6, line 219 Music Program What it is?

#5. Response: Thank you for what the reviewer has pointed out above comments. The subjects selected in our experiment were those of elderly women who wanted to participate in aerobic dance. So the title was decided so on line 244, but the title was corrected to reduce confusion as follow: “2.6. Exercises with EMS Administration” and we corrected the sentence as follows: “The participants in this study were all women who had applied to participate in the aerobic dance class. Most of the elderly participants were obese, and they also had joint disease; therefore, we tried to provide effective exercises for them” on Line 96 to 98 of the new changed manuscript.

#6. Comments and Suggestions: Page 7, line 220 Further, the author lists a number of exercises but again does not indicate a criterion for assessing the intensity of the exercises performed, and especially information on stretching exercises. Author should do so.

#6. Response: Thank you for what the reviewer has pointed out above comments. This sentence would be likely to confuse the reader and modified it as follows: “Upper and lower leg stretching was performed until the participants felt mild discomfort. …” On Line 277 to 288 of the new changed manuscript.

#7. Comments and Suggestions: Page 6, line 230-231 Each 1 MT ...and stored ... and the intensity was adjusted for each individual during aerobic dancing. The intensity of the electrical workout was different from maximal tolerance (1 MT) Author should do so.

#7. Response: Thank you for what the reviewer has pointed out above comments. According to your comments, we changed above sentence as follows: “…The intensity of the electrical workout was different from 1 MT”. On Line 263 of the new changed manuscript.

#8. Comments and Suggestions: How coordination of movements has changed with a combination of voluntary movement plus the evoked contraction during exercise. Have biomechanical motion parameters been recorded? Author should do so.

#8. Response: Thank you for your correction request. According to the investigation in this study, a combination of voluntary movement plus the evoked contraction during exercise tended to decrease by about 30% compared to the original ROM operation. Therefore, we inserted the sentences as follows: “Specifically, according to the investigation in this study, a combination of voluntary movements plus the contractions evoked during exercise tended to decrease by about 30% compared to the original range of movement (ROM).” On Line 290 to 293 of the new changed manuscript.

#9. Comments and Suggestions: And most importantly. If we assume that the results obtained are the influence exerted by an electric training, then what is the mechanism of influence of such a training? Author should do so.

#9. Response: Thank you for what the reviewer has pointed out above comments. According to your suggestion, we inserted the sentences as follows: “Moreover, obese, elderly people are in a situation where a lot of restrictions, such as reduction of ROM, can occur. In this case, the effect of exercise may be reduced due to the inability to effectively cause muscle contractions during exercise. At this time, EMS training induces more muscle contractions in most exercise movements, thereby giving positive effects to the patients” On Line 356 to 360 of the new changed manuscript.

We’ve got the English Editing Service through https://www.mdpi.com/authors/english, again.

Thank you for your comments, we represented the modifications in response to your comments.

March 19, 2020

Reviewer 3 Report

Dear authors,

thank you for the revisions you made to the manuscript.

Unfortunately, you have missed a few essential points which you must absolutely be aware of.

In the following I will take up these points again. Please revise these points before submitting the manuscript again.

  • #2: Sentence sounds very confusing. Rewrite it easier.
  • #7: What kind of controle was done for the water intake?
  • #8: Literature is still missing. Furthermore, you did not point out what the connection betwee the RPE and the CON is. Was the intensity of CON adjusted to the RPE? If no how else? Was the intensity of EMSG determined using the %MT, independent of the RPE scale? 
  • #9: The literature given does not contain any information on the measured creatine levels. Furthermore, the study contents of the study you have listed are fundamentally different, as it mainly used static exercises. In the study presented here, the contents are mainly dynamic. For this reason, the point raised remains, how was this checked? If there was no control, this information has to be in the limitations
  • #11: If you do not use the ANOVA, the alpha error accumulation is still missing. If you have a look at your results in table 4, you would have to adjust the p-values with the 7 measurements you took. Please provide the new results!
  • #12: There is no limitation of the length of the paper in the author guidelines of medicina. Therefore it would be good to discuss the problem of the different outcomes of the studies. Thank you for providing table 3. Sample Size and observation of the diet is still missing in the limitations.

Further aspects:

  • The title sounds confusing now

Kind Regards

Author Response

Answers to reviewer’s comments 

Thank you for your kind advice and comments for publication in Medicina. We re-revised the manuscript as per your comments. We represented the specific modifications in response to the comments by blue-letters in our manuscript. We sincerely appreciate your comments because your comments make our manuscript better.

Reviewer 3:

thank you for the revisions you made to the manuscript. Unfortunately, you have missed a few essential points which you must absolutely be aware of. In the following I will take up these points again. Please revise these points before submitting the manuscript again.

#2. Review: Sentence sounds very confusing. Rewrite it easier:

Comments and Suggestions about the Introduction: If it is not easy for elderly to participate in exercises, why did you choose aerobic dancing?

#2. Response: Thank you for what the reviewer has pointed out above comments. In fact, these authors have served at the Elderly Care Center once a year. The participants in this study were all women, and those who applied to participate in the aerobic dance class. Most of the characteristics of the elderly were obese, and they also had joint disease, so EMS was used to provide effective muscle exercise to them. In view of your comments, we changed and corrected as follows:

“The participants in this study were all women who had applied to participate in the aerobic dance class. Most of the elderly participants were obese, and they also had joint disease; therefore, we tried to provide effective exercises for them” On Line 96 to 98 of the new changed manuscript.

#7. Review: What kind of controle was done for the water intake?

#7. Response: Thank you for what the reviewer has pointed out above comments.

A body composition test can easily use bioelectrical impedance analysis (BIA) to exam the body’s proportion of muscle mass, fat mass, body fluid and so on. The BIA can trace fat mass by conducting high-frequency (500~800 KHz) harmless and calculating the difference of electric resistance between fat tissue and non-fat tissue. However, this BIA method has an error. To reduce this error, food intake was prohibited 4 hours before the test, and water intake was prohibited 1 hour before the test. In particular, urine, which may affect body weight and body water, was urinated 30 minutes before the test.

To avoid confusion, the sentence was corrected as follows:

“To allow accurate inspection, food intake and water intake were prohibited for 4 hours and 1 hour before the test, respectively. Since urine may affect body weight and body fluid, participants urinated 30 minutes before the test [23]” On Line 147 to 149 of the new changed manuscript.

#8. Review: Literature is still missing. Furthermore, you did not point out what the connection betwee the RPE and the CON is. Was the intensity of CON adjusted to the RPE? If no how else? Was the intensity of EMSG determined using the %MT, independent of the RPE scale?

#8. Response: Thank you for what the reviewer has pointed out above comments. In view of your comments, we inserted several sentences and reference as follows:

“The electric stimulation was stopped at the request of the participant when reaching an unbearable level on the RPE scale [29], at which point the intensity was set as 1 MT. In other words, the %MT was obtained through the RPE scale, which is a numerical scale that ranges from 6 to 20, where 6 means "no exertion at all" and 20 means "maximal exertion”. The intensity of exercise was estimated by applying RPE when exercising with music to CON as well as EMSG. The intensity of the electrical workout was different from 1 MT” On Line 258 to 264 of the new changed manuscript.

  1. Borg, G.A. Psychophysical bases of perceived exertion. Med Sci Sports Exerc. 1982, 14, 377-381.

#9. Review: The literature given does not contain any information on the measured creatine (CK) levels. Furthermore, the study contents of the study you have listed are fundamentally different, as it mainly used static exercises. In the study presented here, the contents are mainly dynamic. For this reason, the point raised remains, how was this checked? If there was no control, this information has to be in the limitations.

#9. Response: Thank you for what the reviewer has pointed out above comments. We also measured CK, but we were excluded it from the study's biomarker variable because they seemed to have a distance from the purpose of this study. However, the reviewer's continued argument is added to Table 5 because it indicates that CK is a necessary variable for this study. In view of your comments, we inserted the sentences as follows:

“Creatine kinase (CK) was also included because it is considered to be an indicator of muscle damage during or after exercise with music. CK was analyzed using the Beckmann Coulter Inc. device (Brea, USA) before and after the experiments [25]. The reference range of normal values of CK at rest in a healthy adult is as follows: 52–520 IU/L indicates a high CK level; 35–345 IU/L indicates an intermediate CK level, and 25–145 IU/L indicates a low CK level [26].” On Line 183 to 187 of the new changed manuscript.

#11. Review: If you do not use the ANOVA, the alpha error accumulation is still missing. If you have a look at your results in table 4, you would have to adjust the p-values with the 7 measurements you took. Please provide the new results!

#11. Response: Thank you for what the reviewer has pointed out above comments. According to your opinion, we changed all the results as follows (On Line 305 to 343 of the new changed manuscript)

“3. Results

3.1. Comparison of demographics, calorie intake/output, physical activity, and working capacity

There were no significant differences between groups for all variables, as shown in Table 1. The maximum oxygen uptake of partipants in CON and EMSG were 23.13 ± 4.25 mL/kg/ min and 25.34 ± 6.51 mL/kg/min, respectively, and there was no significant difference between groups. In particular, percentage fat was not significantly different between groups. As shown in Table 3, there were no significant differences between groups in terms of calorie intake, calorie output, and physical activity level for the recorded weeks during the 8-week experimental period.

Table 3. Differences in controlled variables

Items

Week

Groups

ANOVA (p)

CON (n = 12)

EMSG (n = 13)

G

T

G×T

Calorie intake

4

1663.21 ± 124.38

1671.36 ± 182.25

0.819

0.867

0.784

(kcal)

8

1688.69 ± 119.85

1641.43 ± 193.62

Calorie output

4

246.25 ± 51.23

258.08 ± 49.25

0.910

0.442

0.919

(kcal)

8

229.97 ± 52.36

237.88 ± 51.69

PAC

4

1.85 ± 1.45

1.83 ± 1.47

0.862

0.422

0.788

scores

8

1.84 ± 1.63

1.87 ± 1.78

All values are expressed as the mean ± standard deviation. CON, EMSG, and PAC represent the control group, electromyostimulation group, and physical activity category, which were scored by low, moderate, and high activity levels.

3.2. Effect of EMS on Body Composition

As shown in Table 4, no significant effect of the EMS intervention was found for body weight when comparing the intervention and control groups. However, skeletal muscle mass (F = 7.826), fat mass (F = 8.717), percent fat (F = 4.961), and the basal metabolic rate (BMR) (F = 28.770) were significantly different in terms of the group by time interaction. This result shows a significant effect of the EMS intervention concerning body composition.

Table 4. Differences and changes in body composition

Items

Groups

ANOVA (p)

CON (n = 12)

EMSG (n = 13)

G

T

G×T

Body weight

Pre

63.75 ± 3.82

63.05 ± 8.78

0.517

0.010

0.297

(kg)

Post

62.33 ± 4.17

57.54 ± 6.58

Skeletal muscle

Pre

20.46 ± 0.97

20.94 ± 2.03

0.020

0.285

0.010

(kg)

Post

19.82 ± 1.71

22.41 ± 2.19

Fat mass

Pre

25.23 ± 2.93

26.50 ± 5.50

0.649

0.014

0.007

(kg)

Post

25.42 ± 2.66

22.31 ± 3.89

Percent fat

Pre

38.33 ± 4.31

39.21 ± 1.76

0.273

0.466

0.036

(%)

Post

39.66 ± 2.32

34.95 ± 3.82

Basal metabolic rate (kcal)

Pre

1187.00 ± 35.92

1188.69 ± 67.98

0.004

0.249

0.001

Post

1110.17 ± 40.62

1237.77 ± 74.40

All data represent the mean ± standard deviation. CON and EMS mean the control group and electromyostimulation group, respectively.

3.3. Effect of EMS on Biomarkers

As shown in Table 5, the level of IL-6 in CON showed an increasing tendency from Week 0 to Week 8, but this level showed a decreasing tendency in EMSG, although there was no significant difference between groups. However, TNF-α (F = 21.003), CRP (F = 27.825), RSTN (F = 9.520), and CEA (F = 19.331) showed significant differences for the group by time interaction. These positive changes were also represented in the HDL-C and LDL-C measurements. Specifically, although the level of CK in both groups showed an increasing tendency after the experiment, no significant difference was found between groups. These results demonstrate significant effects due to the EMS intervention concerning biomarkers in obese, elderly women.

Table 5. Differences and changes in biomarkers

Items (units)

Groups

ANOVA (p)

CON (n = 12)

EMSG (n = 13)

G

T

G×T

IL-6

Pre

14.29 ± 7.52

14.51 ± 7.14

0.236

0.223

0.051

(pg/mL)

Post

15.41 ± 3.79

9.95 ± 6.31

TNF-a

Pre

28.21 ± 8.51

27.28 ± 12.35

0.039

0.555

0.001

(pg/mL)

Post

36.68 ± 11.68

20.77 ± 8.64

CRP

Pre

34.33 ± 15.80

33.19 ± 10.30

0.001

0.013

0.001

(pg/mL)

Post

54.65 ± 11.66

26.65 ± 8.13

RSTN

Pre

5.92 ± 1.98

5.56 ± 2.55

0.001

0.859

0.005

(ng/mL)

Post

8.43 ± 4.06

3.33 ± 1.09

CEA

Pre

2.12 ± 1.21

2.24 ± 0.66

0.034

0.864

0.001

(ng/mL)

Post

2.88 ± 1.07

1.42 ± 0.19

CK

Pre

222.92 ± 67.13

218.38 ± 51.82

0.869

0.530

0.761

(IU/L)

Post

230.33 ± 80.64

239.69 ± 65.85

HDL-C

Pre

50.92 ± 9.82

47.08 ± 9.99

0.819

0.992

0.008

(mg/dL)

Post

46.33 ± 8.63

51.69 ± 7.70

LDL-C

Pre

134.17 ± 43.47

145.31 ± 35.44

0.332

0.005

0.009

(mg/dL)

Post

131.83 ± 36.60

97.54 ± 23.88

All data represent the mean ± standard deviation. CON, EMSG, IL-6, TNF, CRP, RSTN, CEA, CK, HDL-C, and LDL-C mean the control group, electromyostimulation group, interleukin-6, tumor necrosis factor, C-reactive protein, resistin, carcinoembryonic antigen, creatine kinase, high density lipoprotein-cholesterol, and low density lipoprotein-cholesterol, respectively.

#12. Review: There is no limitation of the length of the paper in the author guidelines of medicina. Therefore it would be good to discuss the problem of the different outcomes of the studies. Thank you for providing table 3. Sample Size and observation of the diet is still missing in the limitations.

#12. Response: Thank you for what the reviewer has pointed out above comments. For avoiding confused sentence, we inserted the sentence as follows:

“5. Conclusions and Limitation

Based on the confirmed homogeneity of this study, the results suggest that the use of progressive electrical EMS impulses for 8 weeks may improve body composition and levels of tumor- or inflammation-related cytokines in elderly, obese women. However, since the IL-6 concentration did not show positive changes in EMSG, we suggest that the use of exercises with music while wearing an EMS suit should be investigated over a longer period or with a greater exercise frequency to determine whether any changes in body composition and biomarkers occur. Although this study showed positive results in terms of body composition and biomarkers, an increased sample size and larger trials are needed to obtain better results.” On Line 450 to 458 of the new changed manuscript.

#Review: The title sounds confusing now.

#Response: Thank you for what the reviewer has pointed out above comments. The title was modified due to instructions from other reviewer.

For avoiding confused sentence, we inserted the sentence as follows: “Exercises with Music While Wearing an EMS Suit Improve Fatness and Biomarkers of Obese Elderly Women”

We’ve got the English Editing Service through https://www.mdpi.com/authors/english again.

Thank you for your comments, we represented the modifications in response to your comments.

March 19, 2020

Round 3

Reviewer 1 Report

1. Page 7 line 279 …..aerobic and anaerobic exercises …. What was an indicator of the aerobic or anaerobic type of exercise performed? Author should do so

The manuscript has improved significantly. However, a significant drawback is still the lack of data on the use of a group of subjects who used exclusively training with electrical stimulation. It is very important. After all, not all elderly patients can actively perform physical exercises, this must be remembered. In this case, electrical stimulation training may be the tool that can stop the lack of motor activity.

Author Response

Answers to reviewer’s comments 

Thank you for your kind advice and comments for publication in Medicina. We re-revised the manuscript as per your comments. We represented the specific modifications in response to the comments by red-letters in our manuscript. We sincerely appreciate your comments because your comments make our manuscript better.

Reviewer 1:

#1. Comments and Suggestions:

  1. Page 7 line 279 …..aerobic and anaerobic exercises …. What was an indicator of the aerobic or anaerobic type of exercise performed? Author should do so

The manuscript has improved significantly. However, a significant drawback is still the lack of data on the use of a group of subjects who used exclusively training with electrical stimulation. It is very important. After all, not all elderly patients can actively perform physical exercises, this must be remembered. In this case, electrical stimulation training may be the tool that can stop the lack of motor activity.

#1. Response:

Thank you for what the reviewer has pointed above a comment, which an indicator of the aerobic exercise was the long time motion during a dance for 40 min and an indicator of the anaerobic exercise was the muscular contraction for 6 second's moving motion from EMS.

We wrote those sentences in our manuscript, “aerobic and anaerobic exercises, which mean the aerobic exercise was the motion for 40 min dance and the anaerobic exercise was the muscular contraction for 6 second's moving motion from EMS,” on Line 270 to Line 272 of the new version of this manuscript.

Re-submission Date

27 March 2020

Reviewer 3 Report

Dear authors,

thank you for the revision of the article and good luck with your future research.

Kind Regards

Author Response

Dear Reviewer 3

Thank you for your kind advice and comments for publication in Medicina.

We re-revised the manuscript as per your comments.

We sincerely appreciate your comments because your comments made our manuscript better.

Best regards,